# Literature Review of *BARD1* as a Cancer Predisposing Gene with a Focus on Breast and Ovarian Cancers

**DOI:** 10.3390/genes11080856

**Published:** 2020-07-27

**Authors:** Wejdan M. Alenezi, Caitlin T. Fierheller, Neil Recio, Patricia N. Tonin

**Affiliations:** 1Department of Human Genetics, McGill University, Montreal, QC H3A 0G4, Canada; wagdan.alenizy@mail.mcgill.ca (W.M.A.); caitlin.fierheller@mail.mcgill.ca (C.T.F.); neil.recio@mail.mcgill.ca (N.R.); 2Cancer Research Program, The Research Institute of the McGill University Health Centre, Montreal, QC H4A 3J1, Canada; 3Department of Medical Laboratory Technology, Taibah University, Medina 42353, Saudi Arabia; 4Department of Medicine, McGill University, Montreal, QC H3A 0G4, Canada

**Keywords:** *BARD1*, cancer predisposing gene, hereditary cancer syndromes, breast cancer, ovarian cancer, next-generation sequencing, multi-gene panel testing

## Abstract

Soon after the discovery of *BRCA1* and *BRCA2* over 20 years ago, it became apparent that not all hereditary breast and/or ovarian cancer syndrome families were explained by germline variants in these cancer predisposing genes, suggesting that other such genes have yet to be discovered. BRCA1-associated ring domain (*BARD1*), a direct interacting partner of BRCA1, was one of the earliest candidates investigated. Sequencing analyses revealed that potentially pathogenic *BARD1* variants likely conferred a low–moderate risk to hereditary breast cancer, but this association is inconsistent. Here, we review studies of *BARD1* as a cancer predisposing gene and illustrate the challenge of discovering additional cancer risk genes for hereditary breast and/or ovarian cancer. We selected peer reviewed research articles that focused on three themes: (i) sequence analyses of *BARD1* to identify potentially pathogenic germline variants in adult hereditary cancer syndromes; (ii) biological assays of *BARD1* variants to assess their effect on protein function; and (iii) association studies of *BARD1* variants in family-based and case-control study groups to assess cancer risk. In conclusion, *BARD1* is likely to be a low–moderate penetrance breast cancer risk gene.

## 1. Introduction

After the discovery of *BRCA1* and *BRCA2*, the highly penetrant breast (BC) and ovarian cancer (OC) predisposing genes, it became evident that about 20% of hereditary BC and/or OC syndrome families did not carry pathogenic variants in these genes. Under the assumption that other genes contribute risk to these cancers, research focused on identifying an additional major risk gene, namely “*BRCAX*”. Since the discovery of *BRCA1* in 1994 [1], 27 cancer predisposing genes that confer significant lifetime risk to hereditary adult cancers have been discovered (reviewed in [2,3]). About half of these genes were discovered using genome-wide linkage analysis to identify candidate loci followed by sequencing candidate genes to identify pathogenic variants segregating in cancer families (reviewed in [2]). However, the rarity of large pedigrees of *BRCA1* and *BRCA2* mutation negative families and availability of family members suitable for linkage analyses posed considerable challenges for gene discovery projects. With knowledge about gene function, sequencing of candidate genes became a favoured strategy for gene discovery. Candidates can be selected based on having similar biological domains, involvement in the same biological pathway or biochemically interacting with known cancer predisposing gene(s). Using this candidate gene approach over 100 genes have been identified of which the majority have failed to be linked with hereditary cancer syndromes, including BC and/or OC, in independent studies (reviewed in [2,4]).

The BRCA1-associated RING domain 1 (*BARD1*) gene was discovered in 1996 in an effort to understand the biological function of BRCA1 protein as shown in the timeline of major discoveries of *BARD1* (Figure 1). BARD1 directly interacts with BRCA1 via its N-terminal RING (really interesting new gene) domain [5]. As pathogenic *BRCA1* variants located in the RING domain disrupt BARD1-BRCA1 interaction, it was hypothesized that variants in *BARD1* may also affect this interaction, thereby making *BARD1* an attractive candidate to pursue in *BRCA1* mutation negative cancer families. Although potentially pathogenic *BARD1* variants have been reported, the role of *BARD1* in cancer predisposition remains inconsistent. Here, we aim to review what is known about *BARD1* as a candidate cancer predisposing gene and discuss challenges facing researchers aiming to discover additional cancer predisposing genes. We focused on publications that: (i) described the analyses of potentially pathogenic germline *BARD1* variants in adult hereditary cancer syndromes; (ii) BARD1 *in vitro* and *in cellulo* assays that assessed the biological effect of potentially pathogenic variants on protein function; and (iii) analyzed cancer risk associated with potentially pathogenic *BARD1* variants in family-based and case-control study groups.

## 2. *BARD1* as a Candidate Cancer Predisposing Gene

Little was known about the biological function of *BRCA1* when it was first reported as a cancer predisposing gene other than the predicted loss-of-function variants suggested that it may behave as tumour suppressor gene [1]. To learn about BRCA1 function, a yeast two-hybrid screening system was used to identify proteins that directly interact with BRCA1 and a protein was identified that interacts directly with its N-terminal RING domain [5]. Repeating the assay using a human cell line, a similar protein which contained a C-terminal motif with significant homology to the conserved C-terminal BRCA1 C terminus (BRCT) domains in BRCA1 and three tandem ankyrin (ANK) domains adjacent to the BRCT domains was identified. The BRCA1–BARD1 interaction was shown to be disrupted by *BRCA1* missense variants [5]. The formation of a stable BRCA1–BARD1 complex was considered to be critical for BRCA1 function and thus BARD1 may play a role as an effector or regulator of BRCA1 [5]. This led to the hypothesis that variants in *BARD1* could encode an aberrant protein affecting the interaction with BRCA1 and predispose to BC and/or OC, especially in hereditary cancer syndrome families not accounted for by *BRCA1* or *BRCA2* [6]. This study was the first report of molecular genetic analyses of *BARD1* in cancer samples in 1998 [6] (Figure 1). In a targeted sequencing analysis of 168 BC, OC, and uterine tumours, the first *BARD1* variant, c.1692G>C; p.Gln564His, was identified in a woman diagnosed with a clear cell histopathological subtype of OC at the age of 73 years [6]. This variant is predicted to change an amino acid residue located near the BRCT domains in BARD1. This individual was also diagnosed with infiltrating lobular BC and six years later with a clear cell endometrial cancer. Sequencing of cloned *BARD1* mRNA transcripts from ovarian and uterine tumour tissues revealed only the variant allele. Sequencing of similarly generated cloned transcripts from non-cancerous uterine tissue revealed both the wild type and variant alleles, suggesting that the missense c.1692G>C; p.Gln564His variant was of germline origin. No family history was available for the variant carrier; thus, it was not possible to determine if this rare variant (not identified in about 300 cancer-free individuals) could be associated with familial cancer. Regardless, the observation that a *BARD1* variant was found in an individual with three primary cancers suggests a strong possibility that it is associated with risk to cancer.

### Subsequent Studies with Inconsistent Results 

The first statistical association was reported by analyzing the *BARD1* c.1670G>C; p.Cys557Ser variant in cancer cases from Finnish [7], Nordic [14], and Icelandic [15] populations versus population-matched cancer-free controls. Collectively, these studies were in agreement with the possibility that carriers of this *BARD1* variant had an increased risk to BC and/or OC. However, a negative association was reported by three different research groups that analyzed case-control groups from Finnish [8], Australian [16], and Polish [17] populations. The inconsistency in results from the same Finnish population was explained by the possibility of a false positive association due to the small sample sizes [7,8]. Indeed, using publicly available population based genetic databases the minor allele frequency (MAF) of *BARD1* c.1670G>C; p.Cys557Ser is 0.015 in the general cancer-free population overall, but it is highly variable among different populations (https://gnomad.broadinstitute.org v2.1.1) [18]. It is the rarest (MAF = 5.2 × 10^−5^) in East Asians and the most common (MAF = 0.024) in the Ashkenazi Jewish population (Appendix A). In the Finnish population, this variant has a frequency of approximately 1%, which is relatively more common compared to known pathogenic variants in well-established BC and OC predisposing genes, *BRCA1* or *BRCA2* (0.001%). This suggests that the effect size of *BARD1* c.1670G>C; p.Cys557Ser could be small, that is, it could confer a low–moderate risk to cancer relative to the high risk associated with *BRCA1/BRCA2* pathogenic variants.

Some subsequent studies reported no statistically significant association of *BARD1* variants with cancer risk, resulting in consideration of *BARD1* not being a strong candidate cancer predisposing gene [10,19,20,21]. In part, early studies lacked statistical power due to limited number of carriers identified in the small size of study groups. In 2017, a positive statistical association of *BARD1* variants with BC was reported in a large sample size of approximately 30,000 BC cases and 30,000 cancer-free controls [11] (Figure 1). Subsequent studies reported consistent positive results and proposed *BARD1* as a candidate cancer predisposing gene where variants confer low–moderate risk for BC, especially for triple-negative BC (TNBC) [13,22,23]. TNBC accounts for 10–20% of all BC cases and is defined by the absence of estrogen and progesterone receptors accompanied by overexpression of human epidermal growth factor receptor 2 (reviewed in [24,25]). These subsequent reports were enriched for TNBC cases, which should have reduced the genetic heterogeneity between BC subtypes [13,22,23,26].

## 3. Potentially Pathogenic Germline *BARD1* Variants Identified in Cancer Cases 

### 3.1. Targeted BARD1 Studies of Breast and Ovarian Cancer

A total of 100 unique germline variants have been reported in studies where targeted analysis of *BARD1* was performed (Appendix A) [6,7,8,9,10,11,12,14,15,17,24,25,26,27,28,29,30,31,32,33,34]. Variants identified in these BC and OC cases include: 12 frameshift; seven nonsense; five splice site; 67 missense; and nine synonymous variants. These variants are rare in the general population, with the MAFs ranging between 4.2 × 10^−7^ and 8.12 × 10^−1^ among cancer-free individuals, though some variants were not identified in this database (https://gnomad.broadinstitute.org v2.1.1) [18]. In *BARD1* targeted sequencing studies that investigated ≥500 cancer cases, the allele frequencies ranged between 0.02%–49% [28,33] and 0.03%–48% [10] in BC and OC cases, respectively. Using the American College of Medical Genetics and Genomics (ACMG) guidelines for variant interpretation, only 19 potentially loss-of-function variants are classified as being pathogenic with a large portion of the remaining variants being classified as a variant of uncertain significance. As the pathogenicity of many of these variants is still unclear, we have re-evaluated all of the identified *BARD1* missense variants using in silico tools that predict the effect on protein function or conservation at that locus. We have found that 29 of the previously reported missense variants are potentially pathogenic based on being predicted to be damaging by at least five out of seven in silico tools and conserved by all three conservation prediction tools (Appendix A). Overall, the mutational spectrum of *BARD1* in BC and/or OC is diverse and there does not appear to be any hotspots in any particular domain (Figure 2A,B).

### 3.2. Multi-Gene Panel and Next-Generation Sequencing Studies Including BARD1 of Breast and Ovarian Cancers

Multi-gene panels that usually contain known high-risk cancer predisposing genes, such as *BRCA1* and *BRCA2*, were used to determine the prevalence and spectrum of variants in the genes in defined study groups for comparative purposes. Germline whole gene sequencing panels included 10–219 genes [16,18,19,20,21,35,36,37,38,39,40,41,42,43,44,45,46,47,48,49,50,51,52,53,54,55,56,57,58,59,60,61,62] and whole exome sequencing strategies analyzed 10–832 genes [63,64,65,66,67]. These sequencing strategies were used in studies on BC and/or OC and male BC, the first being published in 2011 [61] (Figure 1). Collectively, 55 unique variants were reported in familial, high-risk or unselected BC and/or OC cases (Figure 2, Appendix A). Variants identified in these BC and OC cancer cases include: 15 frameshift; 13 nonsense; four splice site; 22 missense; and one synonymous variant. In silico tools revealed that six of the missense variants are predicted to be potentially pathogenic. A germline copy number deletion and deletion of exon 2 were identified in two BC cases [54,65]. A large (1258 bp) heterozygous germline deletion in intron three was identified by multiplex ligation-dependent probe amplification in *BARD1* in a BC case (diagnosed at age 36 years) [68]. This deletion was not detected to have an impact on RNA splicing and allelic imbalance could not be measured as there was no SNP to use. These findings are interesting because these variants may result in partial or complete loss of BARD1 protein. In studies of ≥500 cancer cases, the prevalence of potentially pathogenic *BARD1* variants ranged from 0.18–0.53% [11,47], 0.67–0.9% [13,22], and 0.07–0.23% [37,63] in BC, TNBC and OC cases, respectively. Clearly, in BC and OC cases carriers of *BARD1* variants are much less common than carriers of pathogenic *BRCA1/BRCA2* variants (5–10% each).

### 3.3. BARD1 in Other Cancers

The prevalence of potentially pathogenic germline *BARD1* variants has been investigated in colorectal [32,69], endometrial [70] and pancreatic [71,72] cancers (Figure 2A,B, Appendix A). The carrier frequency of *BARD1* in a familial colorectal cancer study was one in 29 (3.4%) [69]. The carrier frequency of *BARD1* in unselected endometrial cancer was one in 381 (0.26%) [70] and in two pancreatic cancer studies was one in 302 (0.33%) [71] and nine in 96 (9.4%) [72]. The presence of *BARD1* variants in other cancer types suggests that they may also play a role in risk in these diseases. Of the 15 variants that have been identified in these cancer cases, eight have been found in BC and OC cases and two in BC cases. 

### 3.4. Carriers of a BARD1 Variant and a Pathogenic Variant in a Known High-Risk Cancer Predisposing Gene

The carriers of more than one pathogenic variant in different cancer predisposing genes, *BRCA1* and *BRCA2* as an example, has been described but reports are not common. One in 190,000 Europeans is estimated to carry a pathogenic variant in both *BRCA1* and *BRCA2* [73,74,75], and this may be higher in defined populations exhibiting founder effects due to common ancestors, such as in the Ashkenazi Jewish (one in 1800) [76,77] and French Canadian [78] populations. Carriers of a BARD1 variant and a pathogenic *BRCA1* or *BRCA2* variant, that is double heterozygosity, have been reported [14,15,45,48,79].

The contribution of *BARD1* variants to cancer risk or disease progression in carriers of pathogenic variants in known cancer predisposing genes is unknown. Four studies of BC cases have investigated the co-occurrence of carriers of the *BARD1* c.1670G>C; p.Cys557Ser variant with pathogenic variants in *BRCA1* or *BRCA2* [14,15,17,80]. The prevalence of this *BARD1* variant was significantly higher in Icelandic BC cases who also carried the pathogenic *BRCA2* c.771_775del; p.Asn257LysfsTer17 variant known to be a founder pathogenic variant in this population [15]. The risk of BC associated with carrying both variants was estimated to be three-fold higher than carriers of *BRCA2* c.771_775del; p.Asn257LysfsTer17 alone and the lifetime probability of developing BC approaches 100% in carriers of both variants. Three other studies of BC cases from Nordic [14], Polish [17] and mixed populations [80] found that carriers of the *BARD1* c.1670G>C; p.Cys557Ser variant and a *BRCA1* or *BRCA2* pathogenic variant did not confirm this finding. Thus, further studies are needed to clarify the interaction of BARD1 with BRCA1 or BRCA2. It is possible that specific BARD1 variants have additive effects by increasing or modifying risk in a linear or synergistic manner in carriers of other high risk variants (reviewed in [81,82,83]).

In our laboratory, whole exome sequencing of 40 BC cases with pathogenic *BRCA2* variants identified a carrier of *BARD1* c.1339C>G; p.Leu447Val variant (unpublished data). These BC cases were selected by the following criteria: proband with a diagnosis of invasive BC <66 years of age; a strong family history of BC defined by ≥3 BC cases in first-, second- or third-degree relatives in the same familial branch; and all four grandparents originating from Quebec, Canada (French Canadian origin). This rare *BARD1* variant, estimated to occur at an overall frequency of 0.006% in the general cancer-free population (https://gnomad.broadinstitute.org) (Appendix A) [18]. It has been reported twice in the literature and was identified by multi-gene panel testing of 1297 BC cases diagnosed at ≤45 years old [56] and by targeted sequencing of *BARD1* in 4469 familial BC cases [33]. Further analysis of other members of our cancer family showed only one of five other carriers was affected by BC (unpublished data). This case is from the French Canadian population and carries a pathogenic variant in *BRCA2* c.8537_8538delAG; p.Glu2846GlyfsTer22 known to be a founder pathogenic variant as reported by our group and others [84,85,86,87,88,89,90,91,92,93].

## 4. The Biological Effect of Potentially Pathogenic *BARD1* Variants

### 4.1. BARD1 Expression in Mammalian Cells

BARD1 is widely expressed in different types of mammalian cells. Early studies in human cell lines demonstrated a constant level of gene expression throughout the cell cycle, in contrast to BRCA1 expression, which has been shown to increase during late G1 phase and reaches a maximum during S phase [94]. Gene expression studies suggest a role of BARD1 in cell growth. Interestingly, partial repression of mouse Bard1 due to antisense RNAs resulted in development of phenotypes characteristic of the early stages of malignancy in murine mammary epithelial cell lines, suggesting a role of Bard1 in tumourigenesis [95]. Moreover, it has been shown that *BARD1* encodes multiple protein isoforms in human cells that affect cell growth, and the full length protein is able to suppress tumourigenic phenotypes while alternatively spliced transcripts encode proteins that behave as proto-oncogenes (www.genome.ucsc.edu) (reviewed in [96]). Genetic abnormalities affecting chromosome 2q34–q35 region that harbours the *BARD1* locus are not common in BC and OC cases [97,98,99,100]. BARD1 is expressed in breast and ovarian normal tissues as well as in BC and OC tissues (www.proteinatlas.org) and is therefore a plausible candidate cancer predisposing gene.

### 4.2. BARD1 in Animal and Cell Line Models

Genetically engineered mouse models [101,102] suggest that the mouse orthologue of BARD1 plays a role in tumourigenesis. Human BARD1 and mouse Bard1 orthologues have been shown to share about 70% sequence identity [103]. Bard1^−/−^ knockout mice resulted in embryos that died between days E7.5 and E8.5 due to severe impairment of cell proliferation [101]. Bard1 and p53 double knockout embryos displayed increased chromosomal aneuploidy in comparison to p53^−/−^ mice, suggesting a role in maintaining genomic stability [101]. Bard1^+/−^ heterozygous mice were indistinguishable from their wild-type littermates in terms of viability and fertility and did not develop detectable tumours by 21 months of age. Interestingly, conditional knockouts resulting in the inactivation of Bard1 and/or Brca1 resulted in mice that developed BC with histopathologies strikingly similarity to human TNBC, indicating that this is tumourigenic [102]. The authors suggested that this tumourigenic phenotype could be mediated through the BARD1–BRCA1 heterodimer complex.

### 4.3. The Biological Impact of Potentially Pathogenic Variants in Different BARD1 Domains

BARD1 protein (777 amino acids; GenBank Accession number CAE48237.1) is structurally similar to BRCA1 (1,863 amino acids; GenBank Accesstion Number AAC37594.1) by having a single N-terminal RING and two C-terminal BRCT domains. However, BARD1 has four tandem ANK domains adjacent to the BRCT domains, which are absent in BRCA1 [5,6,104]. BARD1 maintains genomic stability by functioning in DNA repair, ubiqutination, and transcriptional regulation (reviewed in [105]). The BRCA1–BARD1 complex functions in the homologous recombination DNA repair pathway that repairs DNA double-strand breaks by assisting in the recruitment of RAD51 to the single stranded DNA as a template for repair and downstream pathway function (reviewed in [105]). Aberrant homologous recombination leaves cells prone to genetic alterations and genome instability.

The RING domain is highly conserved and enriched for cysteine residues that are crucial for binding to zinc cations to ensure proper RING domain folding. The RING domain is crucial for direct BRCA1 interaction and stabilization in a cell cycle-dependent manner and subsequently affects DNA repair [5,94,103,106,107,108,109,110]. RING domains are not only found in proteins that exhibit tumour suppressor or proto-oncogenic function but also in proteins that exhibit E3 ubiquitin ligase activity (reviewed in [111,112]). Studies have shown that ubiquitin ligase activity reaches its maximum when BRCA1 is complexed with the RING domain of BARD1 [110,113]. Studies of germline pathogenic *BRCA1* variants located in the RING domain disrupted the BRCA1–BARD1 interaction [5]. Other *BRCA1* missense variants located in the RING domain impaired E3 ubiquitin ligase activity of the BRCA1–BARD1 heterodimer and affected tumourigenesis [113,114,115]. These findings suggest that having a germline variant in the RING domain of BARD1 could increase risk to BC or OC [6,116]. One frameshift *BARD1* variant was identified in a young age of onset BC case (29 years old at diagnosis) by multi-gene panel testing; this variant is predicted to result in premature termination of the protein and partial loss of the RING domain [46]. The BARD1 c.247A>G; p.Cys83Arg protein isoform retained the ability to interact with BRCA1 and retained E3 ubiquitin ligase activity, but the BARD1–BRCA1 complex was less efficient in binding to the nucleosomes and ubiquitylating histone H2A [117]. In contrast, the BARD1 c.253G>T; p.Val85Leu variant had no effect on the BARD1–BRCA1 interaction and homologous recombination DNA repair activity [118]. We classified both variants as potentially pathogenic based on in silico prediction tools (Figure 2B,C, Appendix A).

Based on amino acid composition BARD1 was thought to contain three ANK repeats but crystallization studies revealed an additional repeat adjacent to the BRCT domain [104]. This newly identified domain may have been missed in earlier studies as it is less conserved than the other ANK domains. These domains are found in a broad spectrum of functionally diverse proteins and have been implicated as sites of highly specific protein-protein interactions (reviewed in [105]). It has been proposed that BARD1 plays a role in apoptosis via its ANK domains in a p53-dependent manner (reviewed in [96,105,119]). Most recently, ANK domains have been shown to be important for recognition of the epigenetic methylation site H4K20me0 on histone H4, for BARD1–BRCA1 recruitment to sites of DNA damage via the homologous recombination DNA repair pathway [120]. Variants in the ANK domains may hinder H4K20me0 recognition which abrogates accumulation of BRCA1 to sites of double-strand breaks, resulting in the activation of alternative DNA repair pathways and genome instability. A deletion of exons 2–6 in *BARD1* results in the loss of most of the RING and ANK domains, and consequently its capacity to interact with BRCA1 [121]. Carriers of 14 frameshift and nine nonsense variants located upstream or within the ANK repeats, which could have the effect of encoding a protein lacking ANK domains, have been reported in BC and OC cases (Appendix A). Interestingly, it was shown that the exonic c.1361C>T; p.Pro454Leu affects the splicing enhancing site, resulting in skipping of exon 5 (Figure 2C) [122], which would result in the loss of a segment of the ANK domains. Studies of *BARD1* missense variants, c.1482A>G; p.Asn470Ser and c.1519G>A; p.Val507Met, suggest that these amino acid substitutions did not significantly affect the secondary structure of BARD1 [104]. Indeed, studies have shown that the *BARD1* c.1519G>A; p.Val507Met protein isoform has no effect on homologous recombination DNA repair pathway activity [118] (Figure 2C, Appendix A).

Both C-terminal BRCT domains are well conserved and are often found in proteins that play a role in cell cycle checkpoint and DNA damage response (reviewed in [105]). BRCT domains in BARD1 are recognized by poly (ADP-ribose) polymerase (PARP) as well as other binding proteins [123,124]. Twenty-four frameshift and 15 nonsense variants in *BARD1* have been reported upstream or within BRCT domains, which could result in losing these domains fully or partially and could affect function (Figure 2, Appendix A). It has been shown that the *BARD1* c. 1793C>T; p.Thr598Ile protein isoform exhibits significantly lower homologous recombination DNA repair activity [118]. This isoform might affect binding to PARP1, as the amino acid change is predicted to alter the protein’s surface and not any of the known binding sites involved in DNA repair [118].

The biological function of *BARD1* variants that lie outside the RING, ANK or BRCT domains have been studied [29,32,125,126,127,128,129,130,131]. The first identified *BARD1* c.1692G>C; p.Gln564His variant and the most prevalent c.1670G>C; p.Cys557Ser variant are located in the linker motif between the ANK and BRCT domains (Figure 2C). The *BARD1* c.1692G>C; p.Gln564His isoform was shown to disrupt the interaction with the Cleavage stimulating factor 50 (CstF50) protein resulting in impairment of RNA processing. Although both of these isoforms showed no effect on the homologous recombination activity [126], they were found to negatively affect the interaction with p53, resulting in impaired p53-dependent apoptosis [29]. These results suggest that the risk to cancer of these variants is unlikely to be a consequence of defects in homologous recombination DNA repair function but may involve cell cycle or RNA processing pathways.

## 5. Risk Assessments for Cancer in Carriers of *BARD1* Variants 

Risk estimates for cancer predisposing genes can be presented as odds ratios (OR) and the risk is interpreted as low (1–1.5), moderate (1.5–5), or high (≥5) [132]. For example, ORs for known BC and OC predisposing genes such as *BRCA1* and *BRCA2* are high, and variants in these genes have been shown to confer high risk to cancer in carriers. Estimates of the lifetime risk for cancer that are associated with carriers of a potentially pathogenic *BARD1* variant have been reported for BC and OC in case-control studies, although most involved familial BC cases. Risk estimates have been determined for individual variants or all variants found in the gene (gene-based risk) (Table 1 and Table 2).

In our literature review, we found three studies that had reported risk estimates for BC, where they were derived from BC cases not selected for family history of cancer [11,22,60] (Table 1). In these studies the ORs for the association of potentially pathogenic *BARD1* variants and risk for BC (any subtype), TNBC or OC were estimated as 2.16 (95% confidence interval (CI) 1.31–3.63, *p* = 2.26 × 10^−3^), 9.76 (95% CI 6.77–13.87, *p* <0.05) and 4.2 (95% CI 1.4–12.5, *p* = 0.02), respectively. The narrow CI for any subtype of BC reported by Couch et al. [11], is a reflection of the large sample size of BC cases (*N* = 28,536) compared with the other studies. In one study the estimated risk was higher for BC to be diagnosed before 40 years of age in familial cases (OR 12.04, 95% CI 5.78–25.08, *p* <0.001) [33], which is consistent with younger age of diagnosis being associated with carriers of pathogenic variants in known cancer predisposing genes, such as *BRCA1* [133]. A meta-analysis of risk estimates for 37 known or proposed genetic risk factors estimated that the OR for *BARD1* was 2.33 for BC and <2 for OC [134].

A number of studies estimated the cancer risk associated with specific variants in *BARD1* in BC and hereditary BC and/or OC syndrome families (Table 2). We found reports describing the risk for nine different *BARD1* variants: one nonsense, one frameshift, six missense and two synonymous variants. Risk estimates are variable across these studies, where the highest risk was observed for c.1690C>T; p.Gln564Ter in association with TNBC cases not selected for family history of cancer (OR 3.62, 95% CI 1.21–10.78, *p* = 0.02). This is consistent with the previous report suggesting that *BARD1* variants may be associated with a higher risk for TNBC [22]. A meta-analysis of the c.1670G>C; p.Cys557Ser variant in 11,870 BC cases and 7687 controls, from Nordic, Australian, Canadian, Russian, English, and Polish populations, concluded that carriers of this *BARD1* variant have a low risk for BC (OR 1.14, 95%CI 0.94–1.34, *p* = 0.13) [135]. While studies estimating the familial risk of *BARD1* variants to OC are still lacking, evidence is emerging that *BARD1* is a potential moderate risk gene for TNBC. 

**Table 1 genes-11-00856-t001:** Risk of breast or ovarian cancer associated with *BARD1*.

Reference	Cancer Type	Selection of Cancer Cases	Number of Cases	Number of Controls	Population	Type of Variants Included	OR (95% CI)	*p*-Value
[11]	Breast	Unselected	28,536	26,078 ^b^	Mixed ^e^	LoF or missense ^f^	2.16 (1.31–3.63)	2.26 × 10^−3^
[52]	Breast	Familial	2134	26,375 ^b^	Mixed ^e^	LoF or missense ^g^	3.18 (1.34–7.36)	1.22 × 10^−2^
[13]	TNBC	High risk	4090	26,079 ^b^	White	LoF or missense ^f^	5.92 (3.36–10.27)	2.20 × 10^−9^
	TNBC	High risk	2003	26,079 ^b^	White	LoF or missense ^f^	4.35 (2.02–9.30)	7.6 × 10^−4^
[23]	Breast	Familial	3667	– ^b^	French	LoF or missense ^g^	2.00 (0.74–4.10)	–
	TNBC	Familial	3667	– ^b^	French	LoF or missense ^g^	11.27 (3.37–25.01)	–
[33]	Breast	Familial	4469	37,265 ^c^	Unknown	LoF or missense ^f^	5.35 (3.17–9.04)	<10^−5^
[22]	TNBC	Unselected	4824	123,136 ^d^	Mixed ^e^	LoF or missense ^f^	9.76 (6.77–13.87)	–
[60]	Ovarian ^a^	Unselected	1915	36,276 ^b^	Mixed ^e^	LoF or missense ^g^	4.2 (1.4–12.5)	0.02
[63]	Ovarian ^a^	High risk	4122	4688	White	LoF	1.59 (0.31–9.56) ^h^	0.57

TNBC = triple-negative breast cancer; LoF = loss-of-function variant (nonsense, frameshift or canonical splice site); OR = odds ratio; 95% CI = 95% confidence interval; ExAC = Exome Aggregation Consortium; FLOSSIES = Fabulous Ladies Over Seventy; gnomAD = Genome Aggregation Database; ACMG = American College of Medical Genetics and Genomics [136,137]; – = not stated; ^a^ Mixed histopathological subtypes; ^b^ ExAC; ^c^ ExAC, FLOSSIES, and geographically matched female controls; ^d^ gnomAD; ^e^ Majority of cases are white; ^f^ ACMG; ^g^ Missense variants were classified using different bioinformatic tools in each study; ^h^ Adjusted OR was calculated as 6.3 (95%CI 0.55–74.25, *p* = 0.19).

**Table 2 genes-11-00856-t002:** Risk of breast or ovarian cancer associated with potentially pathogenic *BARD1* variants.

Reference	Coding DNA Reference Sequence ^a^	Predicted Amino Acid Change	Cancer Type	Selection of Cancer Cases	Number of Cases	Number of Controls	Population	OR (95% CI)	*p*-Value
[28]	c.70C>T	p.Pro24Ser	Breast	Unselected	507	539	China	0.68 (0.52–0.9)	–
[34]	c.722C>G	p.Ser241Cys	Breast	Unselected	143	155	Japan	1.57 (0.61–4.05) ^b^	–
[34]	c.1139_1159del	p.Asn380_Phe387delinsIle	Breast	Unselected	143	155	Japan	0.96 (0.49–1.86) ^c^	–
[34]	c.1134G>C	p.Arg378Ser	Breast	Unselected	143	155	Japan	1.35 (0.7–2.61) ^d^	–
[28]	c.1134G>C	p.Arg378Ser	Breast	Unselected	507	539	China	0.94 (0.72–1.24)	–
[26]	c.1972C>T	p.Arg658Cys	Breast	Unselected	12,476	4707	Poland	1.16 (0.75–1.81)	0.51
[26]	c.1972C>T	p.Arg658Cys	TNBC	Unselected	1120	4707	Poland	0.97 (0.40–2.36)	0.95
[26]	c.1977A>G	p.Arg659=	Breast	Unselected	12,476	4707	Poland	1.32 (0.73–2.40)	0.36
[26]	c.1977A>G	p.Arg659=	TNBC	Unselected	1120	4707	Poland	0.60 (0.14–2.64)	0.5
[34]	c.1518C>T	p.His506=	Breast	Unselected	143	155	Japan	0.71 (0.36–1.42) ^e^	–
[34]	c.1519G>A	p.Val507Met	Breast	Unselected	143	155	Japan	1.28 (0.8–2.04) ^f^	–
[8]	c.1519G>A	p.Val507Met	Breast	Familial	663	718	Finland	1.04 (0.8–1.34)	0.79
[8]	c.1519G>A	p.Val507Met	Breast	Unselected	867	718	Finland	1.27 (0.99–1.63)	0.06
[28]	c.1519G>A	p.Val507Met	Breast	Unselected	507	539	China	0.98 (0.75–1.29)	–
[7]	c.1670G>C	p.Cys557Ser	Breast or ovarian	Familial	126	1018	Finland	4.2 (1.7–10.7)	5 × 10^−3^
[8]	c.1670G>C	p.Cys557Ser	Breast	Familial	926	725	Finland	0.49 (0.24–1.01)	0.05
[8]	c.1670G>C	p.Cys557Ser	Breast	Familial	255	358	Finland	0.70 (0.21–2.34)	0.56
[8]	c.1670G>C	p.Cys557Ser	Breast	Unselected	868	725	Finland	0.74 (0.38–1.44)	0.38
[8]	c.1670G>C	p.Cys557Ser	Breast	Unselected	697	358	Finland	1.09 (0.47–2.56)	0.84
[14]	c.1670G>C	p.Cys557Ser	Breast or ovarian	Familial	757	3591	Nordic	1.9 (1.32–2.83)	10^−3^
[14]	c.1670G>C	p.Cys557Ser	Breast	Unselected	1984	3591	Nordic	1.7 (1.23–2.22)	10^−3^
[15]	c.1670G>C	p.Cys557Ser	Breast	Unselected	992	703	Iceland	1.82 (1.11–3.01)	0.014
[17]	c.1670G>C	p.Cys557Ser	Breast	High risk	3188	1038	Poland	1.2 (0.9–1.7)	0.3
[16]	c.1670G>C	p.Cys557Ser	Breast	Unselected	1136	623	Australia	0.80 (0.50–1.27) ^g^	0.3
[35]	c.1670G>C	p.Cys557Ser	Breast	High risk	322	570	Chile	1.4 (0.6–3.7)	0.47
[64]	c.1690C>T	p.Gln564Ter	Breast	Familial	1018	2036	Poland	1.0 (0.1–8.6)	1
[26]	c.1690C>T	p.Gln564Ter	Breast	Unselected	13,935	5896	Poland, Belarus	2.30 (1.03–5.15) ^h^	0.04
[26]	c.1690C>T	p.Gln564Ter	TNBC	Unselected	1120	4707	Poland	3.62 (1.21–10.78)	0.02

TNBC = triple-negative breast cancer; OR = odds ratio; 95% CI = 95% confidence interval; – = not stated; ^a^ Human GRCh37/hg19 transcript NM_000465.4; OR adjusted for age, first-degree family history of breast cancer, age at first live birth and body mass index was calculated as ^b^ 1.65 (95%CI 0.61–4.43), ^c^ 0.94 (95%CI 0.46–1.90), ^d^ 1.38 (95%CI 0.69–2.76), ^e^ 0.67 (95%CI 0.32–1.41), ^f^ 1.32 (95%CI 0.81–2.16); ^g^ Adjusted OR is shown as the crude OR was not reported; ^h^ OR adjusted for study origin was 2.24 (95%CI 0.99–5.03, *p* = 0.05) and using the Mantel-Haenszel method was 2.12 (95%CI 0.97–4.62, *p* = 0.06).

## 6. *BARD1* in Medical Genetics Settings 

Although risk estimates for developing cancer in carriers of *BARD1* variants remains to be validated for clinical settings, *BARD1* has been included in multi-gene testing panels in hereditary cancer medical genetic settings. Indeed, *BARD1* was added to gene testing panels not long after it was proposed as a cancer predisposing gene. Given that risk for cancer in carriers of *BARD1* variants is equivocal, we questioned if there are recommendations should potentially pathogenic variants be identified by panel testing. We reviewed publicly available resources for statements akin to those providing guidance for the genetic counselling of carriers of known high risk cancer predisposing genes (Table 3). Recommendations for the known high risk genes, *BRCA1* and *BRCA2*, include increased breast awareness and screening, discussion of risk-reducing mastectomy and/or bilateral-salpingo oophorectomy, education about cancer signs and symptoms and genetic counselling (see The National Comprehensive Cancer Network (NCCN) Guidelines for Genetic/Familial High-Risk Assessment: Breast, Ovarian, and Pancreatic Cancer, Version 1.2020). The NCCN guidelines for moderate risk BC genes state that there is an increased risk of female BC with management options for increased screening (mammography), but not risk-reducing mastectomy as there is insufficient evidence. American Society of Clinical Oncology (ASCO) guidelines [138] and the ACMG statement [139] for BC recommend that for moderate-penetrance genes the mutation status should not directly alter treatment, mastectomy or systemic therapy options as data is lacking for these genes. The NCCN Guidelines added *BARD1* to their 2020 version stating that though there is a potential for increased risk for BC in female heterozygous carriers of *BARD1* variants (risk not defined), though there is insufficient evidence to make recommendations (www.nccn.org). The risk for other cancers, including OC, is insufficient to impact risk management. The other resources (ASCO, ACMG, and Institut national d’excellence en santé et en services sociaux (INESSS), a Quebec, Canada resource) did not mention *BARD1*. Nonetheless, *BARD1* continues to be included in multi-gene testing panels for clinical purposes, including commercially available resources. It has been argued that the continued testing of *BARD1* would allow for ready access to carrier status if and when risk estimates and recommendations become available in the future. 

Clinical diagnostic laboratories as well as researchers have used the ClinVar database [141] (www.ncbi.nlm.nih.gov/clinvar) to access reports of variants identified in human diseases, including those for *BARD1*. Submissions to ClinVar are accepted from a number of different groups but the majority come from clinical testing laboratories and very few variants have been assessed for their biological effect on protein function. Each variant is interpreted and assigned a clinical significance: benign, likely benign, uncertain significance, likely pathogenic or pathogenic. There have been a number of reports of *BARD1* variants in the context of hereditary cancer syndromes: 160 pathogenic, 19 pathogenic or likely pathogenic, 56 likely pathogenic, and 1131 variants of uncertain significance (reviewed in [96]). We suggest that ClinVar should be used with caution as a resource to determine if a *BARD1* variant may play a role in hereditary cancer risk, as few of these variants have been vetted by expert review panels akin to those available for variants in *BRCA1* and *BRCA2* (www.brcaexchange.org) [142]. 

The prevalence of variants in high risk cancer predisposing genes in clinical genetic testing settings has been difficult to estimate due to lack of access to data. Recently, Ambry Genetics^®^ (which offers commercially available and clinically approved multi-gene panel tests) developed an open access web-based tool to query prevalence of pathogenic variants in 49 genes, including *BARD1*, found in nine cancer types (www.ambrygen.com/clinician/resources/prevalence-tool) [143]. Pathogenic variants in *BARD1* were identified in 0.27% of BC cases and 0.15% of OC cases (Figure 3, Table 4). Although there is a higher prevalence of carriers in BC cases with a family history of BC and/or OC than those with no family history of cancer, findings are not statistically significant. The prevalence of carriers was also higher in OC cases with a family history of BC than those not selected for family history of cancer, though this difference is not statistically significant. The highest prevalence of carriers was found in TNBC cases (0.85%)—a subtype of BC aforementioned to be most likely to be associated with moderate risk in carriers of *BARD1* variants—which is significantly higher than BC cases unselected for subtype (*p* < 0.001) (Figure 3, Table 4). 

## 7. Our Perspective of *BARD1* as a Cancer Predisposing Gene

Since the discovery over 20 years ago that BARD1 interacts with BRCA1, numerous studies have investigated the possibility that *BARD1* variants may affect risk to the heritable form of BC and OC, and perhaps account for the missing heritability not attributed to *BRCA1/BRCA2*, historically denoted as “*BRCAX*”. However, genetic studies involving family-based cancer cases and case-control study groups have consistently shown that potentially pathogenic variants in *BARD1* are unlikely to: i) confer a high lifetime risk (OR ≥ 5) to BC and OC in heterozygous carriers akin to pathogenic variants described for *BRCA1* and *BRCA2*; and ii) account for the remaining or a significant proportion of heritability of BC and OC attributable to known cancer predisposing genes, especially in cancer syndromes involving these cancer types. Indeed, such studies investigating diverse populations worldwide support the hypothesis that potentially pathogenic variants in *BARD1* confer a low–moderate risk to cancer, with perhaps a moderate risk to TNBC. Molecular biology studies of BARD1 support a role in neoplasia in modifying cell growth and tumour suppressor activity leading to tumourigenesis, particularly in the context of modelling variants found associated with heritable disease. Although collectively the research literature supports a role for *BARD1* variants in the etiology of cancer, there is insufficient evidence to translate findings to clinical settings for the purposes of providing cancer risk in variant carriers or managing cancer care as is currently available for carriers of variants in known high risk genes, such as *BRCA1* and *BRCA2* [144].

There are barriers to estimating risk for *BARD1* variants: i) the rarity of carriers of potentially pathogenic *BARD1* variants; ii) the posited low–moderate risk of cancer associated with *BARD1* variants; and iii) the heterogeneity within and between populations, cancer cases of the same primary tissue site and cancer subtypes (based on histopathology or biomarkers). A family-based study estimated that the relative risk associated with *BARD1* carriers was 2.27 (95%CI 0.47–18.91) for BC [47], suggesting that women with a family history of BC have at least a two-fold increased risk for this disease as compared with those who do not. However, the wide confidence interval, reflecting small sample groups in this study, suggest that this information should be viewed with caution. It will be difficult to identify a sufficiently large sample size to improve risk estimates as the numbers of carriers of *BARD1* variants are rare, as has been observed for carriers of potentially pathogenic *PALB2*, *BRIP1*, *RAD51C* or *RAD51D* variants in hereditary BC and/or OC cancer syndrome families (Figure 3) [145,146]. For *PALB2* the risk of a pathogenic variant was estimated with high confidence in an international study of 524 families [146]. It has been estimated that approximately 40,000 OC cases and 40,000 controls would be required to estimate risk for *BARD1* variant carriers, should this risk be in the low-moderate range [10]. Ideally, the appropriate control population should be matched with the cases on age and sex and have no family history of cancer. Cancer cases should have clearly defined histopathology to more precisely characterize associations with TNBC (Table 1). The design of future studies to assess the risk of *BARD1* pathogenic variants in different cancers will be crucial to determine if translation to patient care in the clinic is appropriate.

A number of *BARD1* missense variants have been characterized and shown to directly alter BARD1 function or its interaction with BRCA1 (Figure 2). It will be important to continue to functionally characterize variants, especially missense variants identified through hereditary cancer studies and clinics, to assess risk. Evaluating new nonsense variants in the C-terminus of the protein will also be important, as previous studies have shown with *BRCA2* c.9976A>T; p.Lys3326Ter that truncation of the end of protein may not significantly increase risk to BC [141,147]. Moreover, *in cellulo* functional assays have demonstrated that the impact of some nonsense variants are not necessarily associated with nonsense-mediated decay [148]. Canonical splicing variants will also be important to assess as aberrant splicing is known to affect the expression of *BARD1* and may lead to mis-localization of the protein (reviewed in [96]). Furthermore, studies are lacking that specifically address genotype-phenotype correlations, variable expressivity, epigenetic modification(s), somatic mosaicism, large genomic rearrangements or de novo germline variants. As *BARD1* variants have also been identified in carriers of known cancer predisposing genes, *BRCA1* and *BRCA2*, it will be important to explore if *BARD1* variants modify risk in an epistatic or additive manner in this context. 

Genetic epidemiology studies have identified a number of host factors that modify cancer risk regardless of variant carrier status (reviewed in [149]). For example, it has been shown that carriers of pathogenic *BRCA1* or *BRCA2* variants have reduced risk for OC due to oral contraceptive pill use or parity or surgical intervention (i.e., tubal ligation or oophorectomy) [150,151,152,153,154]. As the risk attributed to potentially pathogenic *BARD1* variants is expected to be low–moderate, interactions with these host factors should be investigated; results could inform the development of guidelines for patient management and genetic counselling. 

The clinical impact of BARD1 dysfunction in cancer cells is currently unknown. Carriers of pathogenic *BRCA1* or *BRCA2* variants respond well to platinum chemotherapies [155,156] and to new therapies such as PARP inhibitors that take advantage of DNA repair pathways affected by pathogenic variants in these genes [157,158,159,160,161,162]. Assaying *BARD1* variants in women with BC and OC as biomarkers for guiding therapeutic use has not been fully explored.

Evidence from the literature presented in this review is sufficient to conclude that variants in *BARD1* can confer low–moderate risk to BC, especially TNBC, or modify risk to BC in carriers of pathogenic *BRCA1* or *BRCA2* variants (so-called double heterozygous carriers). However, much work remains, and continued research with larger well-defined study subjects with associated clinical correlates, with additionally discovered variants that have been characterized in biological assays, will elucidate the role of BARD1 in the etiology of cancer and help develop guidelines for care in carriers and new therapies. 

## Figures and Tables

**Figure 1 genes-11-00856-f001:**
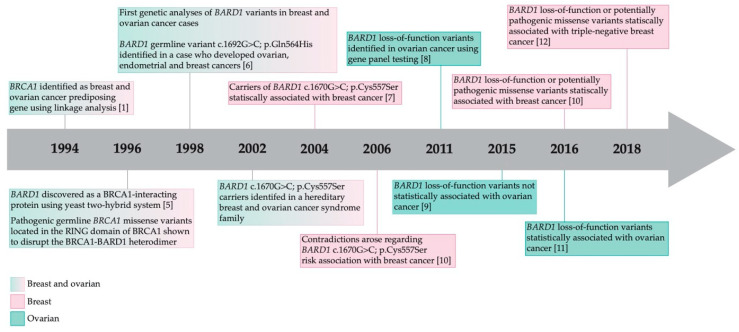
Timeline of the major discoveries associated with *BARD1* function and risk to breast and ovarian cancer [1,5,6,7,8,9,10,11,12,13].

**Figure 2 genes-11-00856-f002:**
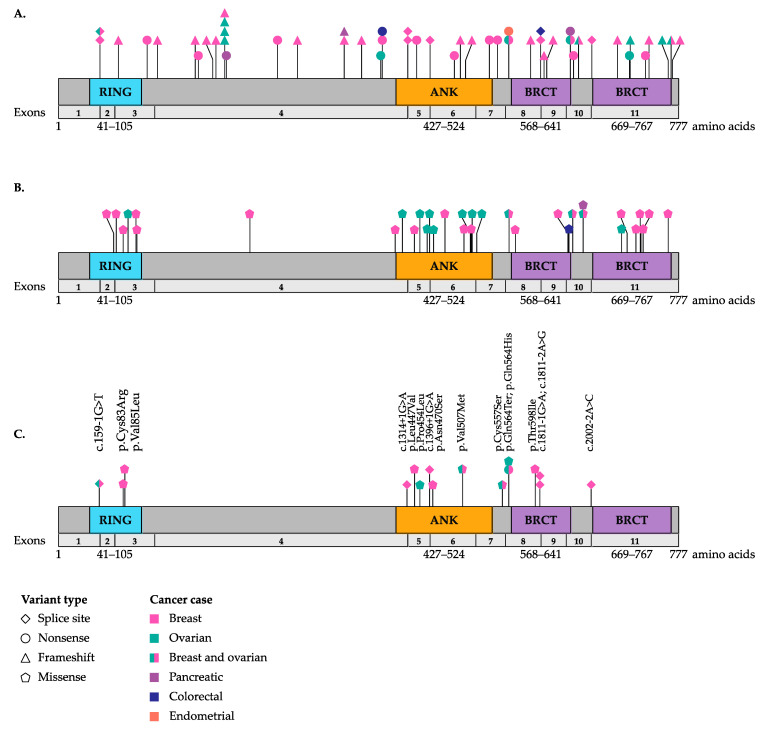
Potentially pathogenic germline variants reported in the literature mapped to full length BARD1 transcript including RING, ANK and BRCT encoding domains. (**A**) Predicted loss-of-function variants including nonsense, frameshift and canonical splice site variants. (**B**) Missense variants predicted to be damaging by in silico tools and conservation tools. (**C**) Variants discussed in the review. RING = Really Interesting New Gene domain; ANK = Ankyrin domain; BRCT = BRCA1 C Terminus domain. See Appendix A for more information.

**Figure 3 genes-11-00856-f003:**
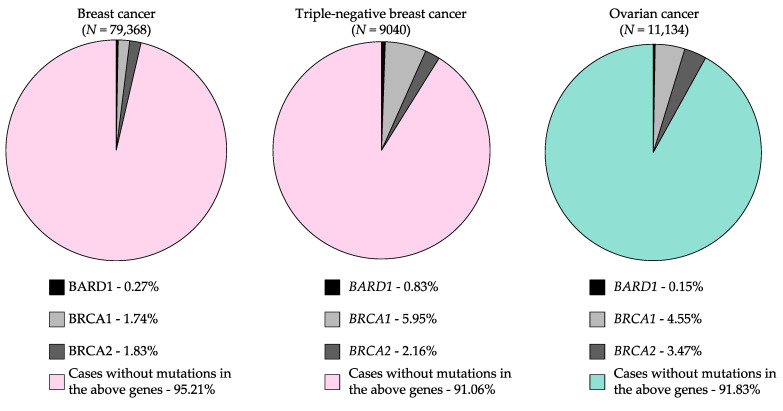
Proportion of carriers of pathogenic *BARD1* variants in breast cancer and ovarian cancer cases relative to carriers of variants in known cancer predisposing genes, *BRCA1* and *BRCA2*. Source: Interactive Prevalence Tables From Multi-Gene Panel Testing: A collaboration between investigators from Mayo Clinic and Ambry Genetics^®^ [143] (www.ambrygen.com/clinician/resources/prevalence-tool).

**Table 3 genes-11-00856-t003:** Publicly available resources reviewed for recommendations for management of carriers of a potentially pathogenic *BARD1* variant.

Reference	Resource Name	Acronym	Resource Type	*BARD1* Included in Resource
www.nccn.org	National Comprehensive Cancer Network	NCCN	Guideline	Yes
[138,140]	American Society of Clinical Oncology	ASCO	Guideline	No
[139]	American College of Medical Genetics and Genomics	ACMG	Statement	No
www.inesss.qc.ca	Institut national d’excellence en santé et en services sociaux	INESSS	Statement	No

**Table 4 genes-11-00856-t004:** Prevalence of *BARD1* variants in breast and ovarian cancer cases ^a^.

Context	Number of Cases Tested	Number of Carriers	%
Personal history of breast cancer	59,375	158	0.27
ER+	30,950	51	0.16
PR+	NA	NA	NA
HER2+	1670	4	0.24
TNBC	6745	56	0.83
Family history of cancer			
Breast	37,372	107	0.29
Ovarian	7990	20	0.25
Breast and ovarian	5138	18	0.35
No cancer	3491	8	0.23
Personal history of ovarian cancer	9149	14	0.15
Family history of cancer			
Breast	3867	9	0.23
Ovarian	NA	NA	NA
Breast and ovarian	NA	NA	NA
No cancer	NA	NA	NA

ER = estrogen receptor; PR = progesterone receptor; HER2 = human epidermal growth factor receptor 2; TNBC = triple-negative breast cancer. ^a^ Source: Interactive Prevalence Tables From Multi-Gene Panel Testing: A collaboration between investigators from Mayo Clinic and Ambry Genetics^®^ [143] (www.ambrygen.com/clinician/resources/prevalence-tool).

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
