# Peer review of "Literature Review of BARD1 as a Cancer Predisposing Gene with a Focus on Breast and Ovarian Cancers"

_genes, 2020, doi:10.3390/genes11080856_

Round 1
Reviewer 1 Report
This is a well written and interesting review summarising the actual knowledge on BARD1 and its role in predisposition to breast and ovarian cancer.
In the presented review, authors summarise essential findings associated with the role of BARD1 in the predisposition to breast/ovarian cancer (Fig. 1), followed by detail description of selected, potentially pathogenic, variants. Furthermore, the authors compile BARD1 experiments using mammalian cells and animal models, risk assessment in carriers of BARD1 alterations, and its significance in genetic counselling. Particularly interesting is the analysis of inconclusive and/or opposite results of questionable c.1670C>T variant.
I would recommend minor revision of the manuscript.
In particular, it would be useful to summarise actual knowledge on large genomic rearrangements identified in BARD1 gene.
Author Response
Dear reviewer,
We are pleased with your overall assessment of our review article.
We agree with your suggestion to include information regarding knowledge of large genomic rearrangements identified in BARD1 gene.
The revised manuscript now includes statements in this regard. We have referenced the one publication that we could find that describes a large deletion in BARD1 (lines 148-151) which also included findings concerning functional consequences. Also, given the paucity of knowledge concerning large structural variants, we also added this among the list of genetic studies for BARD1 that still need to explored in the future (line 476-478).
Reviewer 2 Report
This an outstanding review of the potential for BARD1 as a predisposition gene for breast and ovarian cancers. This comprehensive and well-written review summarizes the history of BARD1 as a candidate cancer predisposition gene and the familial / case-control data as well as functional data evaluating its causality. The authors also delve deeply into the published biological characterization of this protein. This is a quality review that will be of interest to cancer geneticists.
Author Response
Dear Reviewer,
We are pleased with your assessment of our review article.